# Population density and basic reproductive number of COVID-19 across United States counties

**Karla Therese L. Sy**[1,2], **Laura F. White**[3], **Brooke E. Nichols**[2,4,5]*

1 Department of Epidemiology, Boston University School of Public Health, Boston, MA, United States of America, 2 Department of Global Health, Boston University School of Public Health, Boston, MA, United States of America, 3 Department of Biostatistics, Boston University School of Public Health, Boston, MA, United States of America, 4 Health Economics and Epidemiology Research Office, Department of Internal Medicine, School of Clinical Medicine, Faculty of Health Sciences, University of the Witwatersrand, Johannesburg, South Africa, 5 Department of Medical Microbiology, Amsterdam University Medical Center, Amsterdam, The Netherlands

* brooken@bu.edu

**Data Availability Statement:** R code and associated data files are available openly (link: https://figshare.com/articles/dataset/RData_File_-_Data_sets/12858062). Original data files are

## Abstract

The basic reproductive number ($R_0$) is a function of contact rates among individuals, transmission probability, and duration of infectiousness. We sought to determine the association between population density and $R_0$ of SARS-CoV-2 across U.S. counties. We conducted a cross-sectional analysis using linear mixed models with random intercept and fixed slopes to assess the association of population density and $R_0$, and controlled for state-level effects using random intercepts. We also assessed whether the association was differential across county-level main mode of transportation percentage as a proxy for transportation accessibility, and adjusted for median household income. The median $R_0$ among the United States counties was 1.66 (IQR: 1.35–2.11). A population density threshold of 22 people/km$^2$ was needed to sustain an outbreak. Counties with greater population density have greater rates of transmission of SARS-CoV-2, likely due to increased contact rates in areas with greater density. An increase in one unit of log population density increased $R_0$ by 0.16 (95% CI: 0.13 to 0.19). This association remained when adjusted for main mode of transportation and household income. The effect of population density on $R_0$ was not modified by transportation mode. Our findings suggest that dense areas increase contact rates necessary for disease transmission. SARS-CoV-2 $R_0$ estimates need to consider this geographic variability for proper planning and resource allocation, particularly as epidemics newly emerge and old outbreaks resurge.

## Introduction

The COVID-19 pandemic has infected millions of people globally, and there are over 400 thousand reported deaths and 7 million confirmed cases of COVID-19 worldwide [1]. Transmission of airborne and directly transmitted pathogens, such as SARS-CoV-2 (the causative

available publicly (link: https://github.com/nytimes/covid-19-data/blob/master/us-counties.csv).

**Funding:** KTLS and BEN were funded for this work by United States Agency for International Development (USAID) through the following cooperative agreement: AID-OAA-A-15-00070. LFW was supported by NIH R01 GM122876. The funding bodies had no role in the design and conduct of the study; collection, management, analysis, and interpretation of the data; preparation, review, or approval of the manuscript; and decision to submit the manuscript for publication. All authors have seen and approved the manuscript.

**Competing interests:** The authors have declared no conflicts of interest.

agent of COVID-19), have been previously shown to be density-dependent [2–4]. Population density facilitates transmission of disease via close person-to-person contact [5–8], and may support sustained disease transmission due to increased contact rates [9–11]. Large urban areas have more opportunities for disease transmission, and hotspots of SARS-CoV-2 have been mostly concentrated in cities [12].

The basic reproductive number ($R_0$) describes the contagiousness and transmissibility of pathogens, and is a function of contact rates among individuals, transmission probability (probability of transmission per contact), and duration of infectiousness [13]. This is in contrast with the time-varying reproductive number ($R_t$), defined as the number of people in a population who were infected by an infectious individual at a given point in time, which reflects the changing levels of immunity in the population and the impact of control measures limiting transmission [14]. Control measures include implementation of non-pharmaceutical interventions (NPIs), such as face coverings and social distancing. Thus, $R_0$ estimates of COVID-19 are not exclusively determined by the pathogen, and variability in $R_0$ depends on local sociobehavioral and environmental settings, including population density [12].

During the initial phase of the outbreak, or the exponential growth period, we hypothesize that spatial heterogeneity in $R_0$ occurs primarily due to geographic variability in contact rates, since transmission probability and duration of infectiousness remain constant across settings. During this time frame, transmission probabilities across localities are equivalent, because exponential growth occurs prior to the implementation of NPIs which affect likelihood of transmission during contact. Moreover, contact networks are also affected by transportation systems that facilitate disease spread due to increased interconnectivity and mobility between different geographic areas [15, 16]; thus, we also hypothesize that the association of population density and $R_0$ may be differential depending on transportation accessibility, and areas that lack access to efficient modes of transportation would not have the same SARS-CoV-2 growth rate, even in high density areas.

In the current COVID-19 pandemic, evidence of the association of population density and disease transmission have been conflicting [17–20]. The estimation of differential $R_0$ using these area-level factors can assist in more accurate predictions of the rate of spread of SARS-CoV-2 in geographic settings with potential resurgence, where cases have been steadily increasing. In this study, we examine the association of population density with $R_0$ of COVID-19 across United States counties.

## Methods

### Data

We obtained publicly available daily COVID-19 cases among United States counties from the New York Times [21]. For each county, we assumed that the exponential growth period began one week prior to the second daily increase in cases. We assumed that the period of exponential growth lasted approximately 18 days, as previous research have shown the COVID-19 exponential period to be around 20–24 days in New York City [22]. These assumptions calibrated the period of exponential growth accordingly and created reasonable curves that approximated exponential growth across the counties for case data (Fig A in S1 Appendix). The algorithm ensured that the virus had taken hold in the area and allowed a sufficient number of days to estimate the exponential growth rate, as $R_0$ cannot be estimated accurately with sparse data, since it would be uncertain if the county was experiencing a sustained outbreak with community transmission. We restricted our analyses to counties that met a certain threshold of cumulative case counts at the end of the exponential growth period, but including counties with less than 25 cases included daily incidence counts that were insufficient to

calculate $R_0$ and yielded computational errors. Our final analytic sample included counties with 25 cases or greater at the end of the exponential growth period.

Data on the primary mode of transportation to work and median household income were obtained from the most recent 5-year American Community Survey (ACS) 2014–2018 survey estimates from the United States Census Bureau [23]. Main commuting mode was operationalized as the total percentage of people that use private transportation to work, such as those that own private vehicles (car, truck, van, motorcycle) or use a private taxi. Private transportation is a proxy for transportation accessibility in each county, as we hypothesize that areas that that lack efficient modes of transportation may not have as fast of COVID-19 exponential growth, and we want to assess whether the effect of density on $R_0$ would differ due to ease of transportation accessibility. Population and land area were obtained from the 2010 census, and density was calculated by population divided by total square km. All census data were extracted using the R package *tidycensus* [24].

## Statistical analysis

We first compared the densities of counties included in the final analytical sample to those that did not have sufficient case counts with a two-sample Wilcoxon test. $R_0$ was estimated from the exponential growth rate method developed by Wallinga and Lipstich [25] and implemented by the R package *R0* [26], assuming a generation interval with a gamma distribution of mean 4.7 and standard deviation of 2.9 [27]. We then conducted a cross-sectional analysis using linear mixed models with random intercepts for each state and fixed slopes for the counties to assess the association of population density and $R_0$. The linear mixed models allow the intercept to vary among states, which accounts for non-independence among counties within each state, potentially due to variable resource allocation and differing health systems across states.

We also adjusted for county-level main mode of transportation to work percentage and median household income to control for any potential confounding between the association of density and $R_0$. Therefore, we fit 4 models with $R_0$ as the outcome and the following factors as covariates: Model 1: population density; Model 2: population density and the percent of individuals reporting private transportation as their main mode of transportation to work; Model 3: population density and the percent of individuals reporting private transportation, and median household income; Model 4: population density, percent of individuals reporting private transportation, median household income, and the interaction of private transportation use with population density. The associated linear mixed model equation for model 4 is

$$Y_{ij} = \beta_{0j} + \beta_{1j}ln(density\ 1\ unit\ increase)_{ij} + \beta_{2j}\%private\ transportation_{ij}$$
$$+ \beta_{3j}median\ household\ income_{ij} + \beta_{4j}\%private\ transportation_{ij}$$
$$* median\ household\ income_{ij} + e_{ij}$$

where $\beta_{0j} = \gamma_{00} + u_{0j}$, for the *i*-th county for the *j*-th state

## Sensitivity analyses

We conducted three sensitivity analyses to address the limitations of our approach and assess the robustness of our results. First, we conducted a sensitivity analysis using death counts from the New York Times [21] to estimate $R_0$ to limit bias due to differential availability of testing by geographic location. We used the same exponential period as the cases, but with a lag of 14 days to account for the delays from symptom onset to deaths of cases [28–30]. Moreover, the analysis of deaths was restricted to counties with greater than 10 deaths and more than 5 daily

increases in incident deaths, in order to appropriately estimate $R_0$ in counties with sufficient death counts. The daily death data created curves that approximated exponential growth across the counties (Fig B in S1 Appendix). Second, we excluded counties within a radius of 15 miles, the average commuting miles in the United States [31], from counties with densities greater than the 75th percentile. Removing these adjacent counties would demonstrate the extent of biases due to individuals commuting from surrounding counties to cities. If cases are imported from more densely population (i.e. cities) to less dense counties, we could potentially be biasing our estimates downwards. Lastly, we conducted an analysis excluding influential counties with a Cook's distance measure over 4/N for each model. Cook's distance is a commonly used indicator of influence, which measures the extent data points impact the regression parameter estimates [32]. Our sensitivity analysis excluding influential counties ensures that our findings were not driven by these highly influential observations, and that the association holds for other counties.

All analyses was conducted in R version 4.0.0 [33]. The figure and removal of adjacent counties in the sensitivity analyses were done with ArcGIS [34].

## Results

The United States has 3,221 counties and county equivalents. When restricting to counties with greater than 25 cases, 1,151 (35.73%) counties were included (Fig 1). The median $R_0$ among the counties was 1.66 (IQR: 1.35–2.11). The median start and end of the exponential growth period was March 25, 2020 and April 9, 2020 respectively, and differed for each county depending on the start of disease transmission. The median density in counties included and not included in the analysis were 53.8 people/km$^2$ (IQR: 21.24–144.05) and 11.3 people/km$^2$ (IQR: 3. 56–23.88) respectively, and the difference was statistically significant (p <0.0001). A

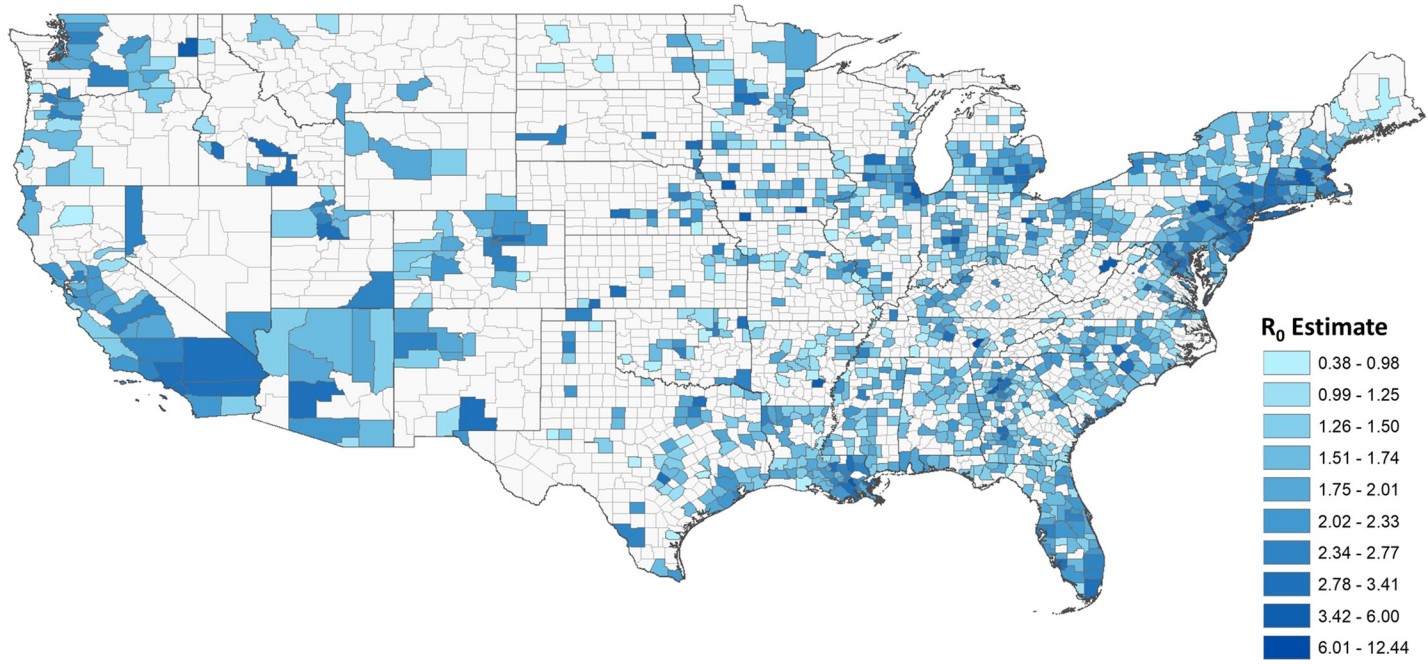

**Fig 1. Basic reproductive number (R$_0$) estimates across United States counties.** Larger R$_0$ indicates greater transmission during the initial phase of the outbreak, or the exponential growth period. We restricted calculation of R$_0$ to counties with greater than 25 cases at the end of the exponential growth period (n = 1,151), as R$_0$ cannot be estimated accurately with sparse data and it would be uncertain if the county was experiencing a sustained outbreak with community transmission.

population density threshold of approximately 22 people/km$^2$ was needed to sustain an outbreak (Fig 2).

An increase in one unit of log population density increased $R_0$ by 0.16 (95% CI = 0.13 to 0.19) (Model 1; Table 1), or the doubling of population density increased the $R_0$ on average by 0.11 (95% CI = 0.09 to 0.13). When adjusted for percent of private transportation and median household income, the association of log population density and $R_0$ remained unchanged (Model 3; Table 1). There was no significant interaction, and the effect of population density on $R_0$ was the same among counties with a larger percentage with private vehicles as their transportation to work, but was no longer significant (Model 4; Table 1). Therefore, model 4 should be not be interpreted, since the interaction does not contribute to the model and only serves to decrease the precision (larger confidence intervals) of the measures of association of interest. $R_0$ decreased by 0.12 (95% CI = -0.02 to -0.04) with an 10% increase in private transportation as the main commute mode, accounting for population density and median household income (Model 3; Table 1).

In all three sensitivity analyses, population density remained positively associated with $R_0$, demonstrating the robustness of our main analysis. First, death data was used to calculate $R_0$ from 301 counties. The median $R_0$ among the counties that had sufficient death counts was 1.40 (IQR: 1.05–1.78). The unadjusted association between population density and $R_0$ remained consistent ($\beta$ = 0.18, 95% CI = 0.14 to 0.23) (Model 1a; Table 2), and there were no significant interactions (Model 4a; Table 2). Next, there were 288 counties above the 75th percentile, and 414 counties that were within 15 miles of these counties high-density counties. We removed these 414 counties from the sample, and using the subsample of 737 counties, our findings remained consistent (Table 2). Influential counties were also not driving the

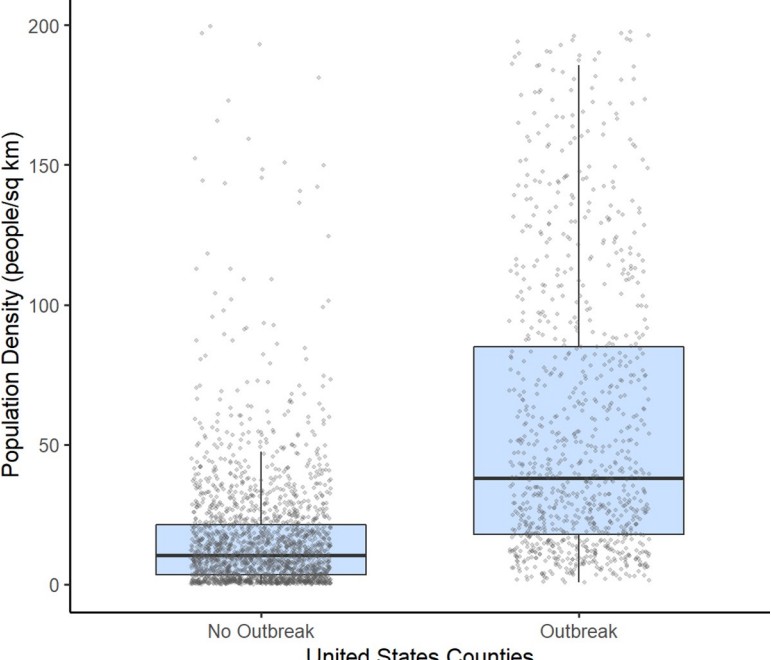

**Fig 2. Population density threshold required to establish a sustained outbreak in United States counties.** A population density of approximately 22 people/km$^2$ was needed to sustain an outbreak, which is approximately equal to the lower IQR of the counties with established COVID-19 outbreaks [median = 53.8 population/km$^2$; IQR = 21.24, 144.05], and a slightly less than the upper IQR of the counties with no COVID-19 outbreaks [median = 11.3 population/km$^2$; IQR = 3.56, 23.88]. The grey circles represent the individual county population densities.

**Table 1. Linear mixed models (random intercept, fixed slope) evaluating the association between population density and basic reproductive number ($R_0$) among United States counties.**

| | Model 1 | Model 2 | Model 3 | Model 4 |
|---|---|---|---|---|
| Log of population density | **0.16*** (0.13–0.19)** | **0.15*** (0.12, 0.18)** | **0.15*** (0.12, 0.18)** | 0.20 (-0.06, 0.47) |
| Percent of private transportation (10%) | | **-0.12*** (-0.19, -0.04)** | **-0.12** (-0.2, -0.04)** | -0.08 (-0.27, 0.11) |
| Median household income ($10,000) | | | 0.00 (-0.03, 0.03) | 0.00 (-0.03, 0.03) |
| Interaction of population density and transportation | | | | -0.01 (-0.04, 0.02) |

pvalue

* < 0.05

** < 0.01

*** < 0.001.

Model 1- unadjusted association of population density and $R_0$.

Model 2 –association of population density adjusted for the percent of individuals reporting private transportation.

Model 3 –association of population density, percent of individuals reporting private transportation, and median household income.

Model 4 –association of population density, percent of individuals reporting private transportation, median household income, and the interaction of population density and transportation.

Estimates for each model is a slope (beta) with a null of 0; a positive slope indicates that an increase in the log of population density increases $R_0$ by the beta estimate for the log of population density. The interaction term indicates that the association of population density and $R_0$ differs depending on the percentage of people using the private transportation for work.

**Table 2. Sensitivity analysis of linear mixed models (random intercept, fixed slope) using (a) deaths only, (b) removing counties within 15 miles of high-density counties, and (c) removing high influence counties.**

| | Model 1 | Model 2 | Model 3 | Model 4 |
|---|---|---|---|---|
| **Deaths only** | | | | |
| Log of population density | **0.18*** (0.14, 0.23)** | **0.14*** (0.09, 0.19)** | **0.12*** (0.07, 0.17)** | **0.48* (0.01, 0.94)** |
| Percent of private transportation (10%) | | **-0.16** (-0.26, -0.06)** | **-0.15** (-0.24, -0.05)** | 0.14 (-0.24, 0.53) |
| Median household income ($10,000) | | | **0.06** (0.03, 0.1)** | **0.07*** (0.03, 0.11)** |
| Interaction of population density and transportation | | | | -0.04 (-0.09, 0.01) |
| **No counties adjacent to high density counties** | | | | |
| Log of population density | **0.17*** (0.14, 0.2)** | **0.17*** (0.13, 0.2)** | **0.16*** (0.13, 0.2)** | 0.02 (-0.13, 0.17) |
| Percent of private transportation (10%) | | 0.01 (-0.07, 0.1) | -0.01 (-0.12, 0.1) | -0.12 (-0.28, 0.03) |
| Median household income ($10,000) | | | 0.02 (-0.03, 0.06) | 0.01 (-0.03, 0.05) |
| Interaction of population density and transportation | | | | 0.03 (0, 0.06) |
| **Removing high influence counties** | | | | |
| Log of population density | **0.18*** (0.15, 0.2)** | **0.18*** (0.16, 0.21)** | **0.18*** (0.15, 0.2)** | **0.38* (0.04, 0.72)** |
| Percent of private transportation (10%) | | -0.01 (-0.09, 0.07) | -0.02 (-0.1, 0.07) | 0.09 (-0.1, 0.29) |
| Median household income ($10,000) | | | 0 (-0.03, 0.02) | 0 (-0.02, 0.03) |
| Interaction of population density and transportation | | | | -0.02 (-0.06, 0.02) |

pvalue

* < 0.05

** < 0.01

*** < 0.001.

Model 1– unadjusted association of population density and $R_0$.

Model 2 –association of population density adjusted for the percent of individuals reporting private transportation.

Model 3 –association of population density, percent of individuals reporting private transportation, and median household income.

association of population density and $R_0$, and our results remained robust (Table 2). For the two sensitivity analyses excluding counties adjacent to highly dense counties and excluding high influence counties, however, the association of private transportation usage and $R_0$ did not remain (Models 2a, 3a, 2b, 3b; Table 2).

## Discussion

Our findings show that the basic reproductive number ($R_0$) is associated with population density, even when percent of individuals that use private transportation and median income were accounted for. In these settings, greater population density may potentially facilitate interactions between susceptible and infectious individuals in densely-populated networks, which sustain continued transmission and spread of COVID-19. Moreover, we see that population density continues to have an important impact on disease transmission regardless of transportation accessibility and median income, suggesting that the opportunity for effective contacts are mostly driven by crowding in denser areas, increasing the contact rates necessary for disease spread. However, we did not see that density-dependence is differential across transportation accessibility, as demonstrated by the non-significant interaction of population density and transportation. Our findings are consistent with previous research that have demonstrated a strong relationship between population density and other infectious diseases [4, 6, 7, 9, 35]. In the current SARS-CoV-2 pandemic, recent literature has been conflicting, where some research also suggests a density-dependence of COVID-19 transmission [17, 36] and other measures of the severity of the outbreak [19, 37], while other research suggests that there are other factors that can better explain the pandemic [18, 38]. However, to our knowledge, our results are one of the first to show that population density is an important driver of COVID-19 transmission, even in areas where residents rely more on private modes of transportation. Moreover, even though transmission is less in lower density areas (i.e. rural areas), rural settings may eventually disproportionately be more vulnerable to COVID-19 morbidity and mortality. Individuals in rural areas are generally older, have more underlying conditions, have less access to care, and have fewer ICU beds, ventilators, and facilities needed for severe COVID-19 treatment [39–42]. Further research is needed on the overall burden of COVID-19 across the spectrum of population density.

Geographic estimates of $R_0$ of SARS-CoV-2 need to take into account the specific area's population density, since the $R_0$ estimate is dependent on both the pathogenicity of the virus as well as environmental influences. In countries where cases are only on the starting to climb, such as countries in Latin America and Southern Africa [1, 43], or there is a resurgence of cases, such as India, Iraq, and Israel [44], area-specific density can assist in predictions of $R_0$, which is important because epidemiological forecasts and predictive models are sensitive to small changes in $R_0$ inputs. Accurate estimation of $R_0$ consequently leads to more precise approximations of the epidemic size, so that governments can appropriately allocate resources and coordinate mitigation strategies. Moreover, as cities and states reopen in the United States, and if there is a second-wave of infections, areas with higher density accessibility will likely have greater SARS-CoV-2 resurgence.

Our study has a number of limitations. While we demonstrate that population density is associated with $R_0$, the estimation of $R_0$ can be biased depending on the data and assumptions adopted. However, our main aim in this analysis was to evaluate the association between population density and $R_0$, and not to accurately estimate $R_0$. Thus, any biases in estimation of $R_0$ due to underlying assumptions would likely be non-differential across counties, and would still yield similar results. In addition, we estimated $R_0$ based on the number of reported cases; therefore, the incidence of COVID-19 across US counties may be underestimated at varying rates due to differential testing. Testing data at the county-level currently do not exist, and we

were unable to adjust for the number of tests performed. Confounding of true epidemic growth by increase in testing could also be a potential constraint to the robustness of the analysis. To mitigate this limitation, we included a random intercept term to adjust for state-level effects, and thus differential testing across states were accounted by our model. Differential testing by local governments within states are less likely to strongly impact our findings, as most funding and budgets for COVID-19 is distributed at the state-level [45, 46]. We also conducted a sensitivity analysis using death data which demonstrates the robustness of our findings. Furthermore, we utilized a number of assumptions based on previous findings to calibrate the exponential growth period, which ensured that the virus had taken hold and allowed a sufficient number of days and case counts to estimate exponential growth. There are potential for biases in our method; for example, there is the possibility that some NPIs were introduced in the initial outbreak stage of COVID-19 in some counties; however, if this was the case, then case counts and subsequently $R_0$ would even higher than we calculated, and thus our associations of density and $R_0$ was underestimated. However, we implemented numerous ways to limit the biases. The exponential growth period was restricted to approximately 14 days at the start of the epidemic, where we would expect limited increases in testing and thus would not affect $R_0$ substantially. Moreover, we plotted the calibrated exponential growth curves of all the counties included in our analysis, which gave us reasonable curves that approximated exponential growth for case and death data. Another limitation is that we had to only include counties that had sufficient case data in order to accurately estimate $R_0$; however, if we included all counties, the true association between population density and $R_0$ would likely be greater than what we report in our analysis given our findings that the counties excluded in the analysis had a significantly lower density and expected very low $R_0$ due to lack of cases. Another limitation is that our model also assumes homogenous mixing, which may can be an oversimplification of the heterogeneity in contact patterns within populations [4, 47]. However, previous research has shown that population structure only changes $R_0$ estimates slightly [48], and assumptions of well-mixed populations are valid in small-to-medium spatial scales [17]. Moreover, our method loses spatial granularity in assessing $R_0$ in counties, especially in counties with spatially heterogenous clustering. The aim of our study, however, was to provide a generalizable estimate of the association between population density and $R_0$, in order to appropriately estimate potential for disease transmission, rather than a microspatial estimate that may not be generalizable to other settings. Finally, an important confounder that we were unable to adjust for is the number of importations of SARS-CoV-2 in these counties, as more urbanized areas are more likely to have links with countries and other states where the virus could have originated from. Even so, we still see that once an area is seeded with COVID-19, the growth rate is greater in denser areas during the time period prior to implementation of NPIs.

## Conclusions

Counties with greater population density have greater rates of transmission of SARS-CoV-2, likely due to increased contact rates in areas with greater population density. Population density affects the network of contacts necessary for disease transmission, and SARS-CoV-2 $R_0$ estimates need to consider this variability for proper planning and resource allocation, particularly as epidemics newly emerge and old outbreaks resurge.

## Supporting information

**S1 Appendix. Population density and basic reproductive number of COVID-19 across United States counties.**
(DOCX)

## Acknowledgments

**Disclaimers:** The author's views expressed in this publication do not necessarily reflect the views of the United States Agency for International Development or the United States Government.

## Author Contributions

**Conceptualization:** Karla Therese L. Sy, Brooke E. Nichols.

**Data curation:** Karla Therese L. Sy.

**Formal analysis:** Karla Therese L. Sy.

**Funding acquisition:** Brooke E. Nichols.

**Investigation:** Brooke E. Nichols.

**Methodology:** Karla Therese L. Sy, Laura F. White, Brooke E. Nichols.

**Supervision:** Brooke E. Nichols.

**Validation:** Karla Therese L. Sy.

**Visualization:** Karla Therese L. Sy.

**Writing – original draft:** Karla Therese L. Sy.

**Writing – review & editing:** Karla Therese L. Sy, Laura F. White, Brooke E. Nichols.

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
