## [Decision Letter · Decision Letter 0]

2 Feb 2021

PONE-D-20-40744

Population density and basic reproductive number of COVID-19 across United States counties

PLOS ONE

Dear Dr. Nichols,

Thank you for submitting your manuscript to PLOS ONE. After careful consideration, we feel that it has merit but does not fully meet PLOS ONE’s publication criteria as it currently stands. Therefore, we invite you to submit a revised version of the manuscript that addresses the points raised during the review process.

We look forward to receiving your revised manuscript.

Kind regards,

Martial L Ndeffo Mbah, Ph.D

Academic Editor

PLOS ONE

Additional Editor Comments (if provided):

Reviewer #2 has very constructive comments which should help greatly improve the quality of the manuscript and robustness of its results.

Journal Requirements

Reviewers' comments:

Reviewer's Responses to Questions

**Comments to the Author**

1. Is the manuscript technically sound, and do the data support the conclusions?

Reviewer #1: Yes

Reviewer #2: Partly

2. Has the statistical analysis been performed appropriately and rigorously? 

Reviewer #1: Yes

Reviewer #2: Yes

3. Have the authors made all data underlying the findings in their manuscript fully available?

Reviewer #1: Yes

Reviewer #2: Yes

4. Is the manuscript presented in an intelligible fashion and written in standard English?

Reviewer #1: Yes

Reviewer #2: Yes

5. Review Comments to the Author

Reviewer #1: This paper is highly informative and timely. My main concern was about differential testing across counties during the exponential growth period, but the authors acknowledge this limitation and do their best to mitigate with random intercept terms. Aside from this, please note a few suggestions/points of clarification below.

• Some brief text on the difference between R0 versus R(t) might be helpful.

• Instead of saying “22 population/km2” perhaps say “22 people/km2?”

• Lines 62 and 63 read strangely, consider rephrasing.

• Consider changing “case” to “cases” in line 76.

• I have trouble understanding line 77. Does it mean to say the exponential growth period began one week prior to the second daily increase in cases?

• Some more explanation about “influential counties” around lines 131-133 would be helpful.

• Consider changing “densely-population” to “densely-populated” in line 196.

Reviewer #2: Comments

This study examined whether cross-county variation in R0 could be explained by variation in population density. The results indicated that an increase in population density increased R0. The method used was linear mixed models with random intercept and fixed slopes. A merit of this study is that it utilized subnational data that contained rich information on local settings. The following are my suggestions for making improvement of the manuscript.

1. The basic reproductive number has many limitations. It must be estimated and varies with the mathematical model and assumptions adopted. R0 was rarely observed in the real world. Authors may consider using other measures of COVID-19 transmission that are more straightforward, such as case number or doubling/halving time, as sensitivity analysis or alternative models. Analysis of those measures may better serve public policy.

2. Literature review should be elaborated, especially studies relating to population density and disease transmission.

3. It is not clear how the period of exponential growth was calibrated, and the reason for assuming the exponential growth period was one week prior to the second daily increases in cases.

4. It is not clear how mixed linear models fit into the cross-sectional data structure, as those models are usually for multilevel/hierarchical, longitudinal, or correlated data. Does each county have repeated measurements? Are measurements made on clusters?

5. Figures seem to be poorly presented.

6. It was possible that some NPIs were introduced in the initial outbreak stage of COVID-19 in some counties, such as public information campaigns. This would make the study assumption invalid.

7. Authors are advised to provide explanations for insignificance of the coefficient estimate of population density in Model 4.

6. PLOS authors have the option to publish the peer review history of their article (what does this mean?). If published, this will include your full peer review and any attached files.

Reviewer #1: No

Reviewer #2: No

---

## [Author Response · Author response to Decision Letter 0]

23 Feb 2021

Response to Reviewers

Reviewer #1

This paper is highly informative and timely. My main concern was about differential testing across counties during the exponential growth period, but the authors acknowledge this limitation and do their best to mitigate with random intercept terms. Aside from this, please note a few suggestions/points of clarification below.

Comment 1

Some brief text on the difference between R0 versus R(t) might be helpful.

Response 1

We thank the reviewer for the positive and helpful assessment of the manuscript. We agree with the reviewer that briefly describing the difference between these two measures would add additional context. On page 3, paragraph 2, we added the following text:

“This is in contrast with the time-varying reproductive number (Rt), defined as the number of people in a population who were infected by an infectious individual at a given point in time, which reflects the changing levels of immunity in the population and the impact of control measures limiting transmission.”

Comment 2

• Instead of saying “22 population/km2” perhaps say “22 people/km2?”

Response 2

We thank the reviewer for the suggestion, and have changed the text accordingly. We have also changed the text of Figure 2 for consistency.

Comment 3

• Lines 62 and 63 read strangely, consider rephrasing.

Response 3

We thank the reviewer for the thoughtful suggestion. We rephrased both this sentence and the sentence before it (now lines 64 to 66), in order to improve the transition from the previous sentence.

“During the initial phase of the outbreak, or the exponential growth period, we hypothesize that spatial heterogeneity in R0 occurs primarily due to geographic variability in contact rates, since transmission probability and population size remain constant across settings. During this time frame, transmission probabilities across localities are equivalent, because exponential growth occurs prior to the implementation of NPIs which affect likelihood of transmission during contact.”

Comment 4

• Consider changing “case” to “cases” in line 76.

Response 4

We have modified the text from “case” to “cases” in line 76.

Comment 5

• I have trouble understanding line 77. Does it mean to say the exponential growth period began one week prior to the second daily increase in cases?

Response 5

We thank the reviewer for realizing our typographical error. Yes, the reviewer is right, we meant to say that “we assumed that the exponential growth period began one week prior to the second daily increase in cases”. We have changed the text accordingly (now line 82). In the same paragraph, we state that “The algorithm ensured that the virus had taken hold in the area and allowed a sufficient number of days to estimate the exponential growth rate, as R0 cannot be estimated accurately with sparse data, since it would be uncertain if the county was experiencing a sustained outbreak with community transmission.” 

Comment 6

• Some more explanation about “influential counties” around lines 131-133 would be helpful.

Response 6

We thank the reviewer, and have added additional information and a citation on “influential counties”, Cook’s D, and influence (now lines 142 to 146).

“Lastly, we conducted an analysis excluding influential counties with a Cook’s distance measure over 4/N for each model. Cook’s distance is a commonly used indicator of influence, which measures the extent data points impact the regression parameter estimates [31]. Our sensitivity analysis excluding influential counties ensures that our findings were not driven by these highly influential observations, and that the association holds for other counties.”

Comment 7

• Consider changing “densely-population” to “densely-populated” in line 196.

Response 7

We thank the review for noticing this typographical error, and have changed “densely-population” to “densely-populated” in line 196 (now line 213).

Reviewer #2

This study examined whether cross-county variation in R0 could be explained by variation in population density. The results indicated that an increase in population density increased R0. The method used was linear mixed models with random intercept and fixed slopes. A merit of this study is that it utilized subnational data that contained rich information on local settings. The following are my suggestions for making improvement of the manuscript.

Comment 1

1. The basic reproductive number has many limitations. It must be estimated and varies with the mathematical model and assumptions adopted. R0 was rarely observed in the real world. Authors may consider using other measures of COVID-19 transmission that are more straightforward, such as case number or doubling/halving time, as sensitivity analysis or alternative models. Analysis of those measures may better serve public policy.

Response 1

We thank the reviewer for the positive assessment of our manuscript and suggestions for improvement. We agree that the estimation of the basic reproductive number can be biased depending on the data and assumptions adopted. However, our main aim in this analysis was to evaluate the association between population density and R0, and not to accurately estimate R0. Thus, any biases in estimation of R0 due these underlying assumptions would likely be non-differential across counties, and would yield similar results to our findings. We further outline these limitations in the Discussion section (p. 242)

“While we demonstrate that population density is associated with R0, the estimation of R0 can be biased depending on the data and assumptions adopted. However, our main aim in this analysis was to evaluate the association between population density and R0, and not to accurately estimate R0. Thus, any biases in estimation of R0 due to underlying assumptions would likely be non-differential across counties, and would still yield similar results. In addition, we estimated R0 based on the number of reported cases; therefore, the incidence of COVID-19 across US counties may be underestimated at varying rates due to differential testing. Testing data at the county-level currently do not exist, and we were unable to adjust for the number of tests performed. Confounding of true epidemic growth by increase in testing could also be a potential constraint to the robustness of the analysis.”

We also outlined numerous analytics methods and sensitivity analyses to mitigate these limitations (p. 250).

“To mitigate this limitation, we included a random intercept term to adjust for state-level effects, and thus differential testing across states were accounted by our model. Differential testing by local governments within states are less likely to strongly impact our findings, as most funding and budgets for COVID-19 is distributed at the state-level [45, 46]. We also conducted a sensitivity analysis using death data which demonstrates the robustness of our findings.”

As our main aim in this study is the association of population density and transmission of SARS-CoV-2 as measured by R0, we believe that using a different a measure of COVID-19 severity such as case counts or doubling/halving time would ask a different research question, and measures may still not be free of biases. Evaluating the association of population density and other COVID-19 measures is important, but is not a focus of this paper. In both the Introduction (p. 3) and the Discussion section (p. 12), we reiterate why we chose to focus on R0 as the main outcome.

“During the initial phase of the outbreak, or the exponential growth period, we hypothesize that spatial heterogeneity in R0 occurs primarily due to geographic variability in contact rates, since transmission probability and population size remain constant across settings.”

“Geographic estimates of R0 of SARS-CoV-2 need to take into account the specific area’s population density, since the R0 estimate is dependent on both the pathogenicity of the virus as well as environmental influences… Accurate estimation of R0 consequently leads to more precise approximations of the epidemic size, so that governments can appropriately allocate resources and coordinate mitigation strategies.”

Comment 2

2. Literature review should be elaborated, especially studies relating to population density and disease transmission.

Response 2

We appreciate the helpful suggestion by the reviewer. We have added additional literature review outlining studies relating population density and disease transmission of other infectious diseases. Moreover, we have updated the literature review on studies specifically relating population density and COVID-19 transmission (p. 12).

Our findings are consistent with previous research that have demonstrated a strong relationship between population density and other infectious diseases [4, 6, 7, 9, 35]. In the current SARS-CoV-2 pandemic, recent literature has been conflicting, where some research also suggests a density-dependence of COVID-19 transmission [17, 36] and other measures of the severity of the outbreak [19, 37], while other research suggests that there are other factors that can better explain the pandemic [18, 38]. However, to our knowledge, our results are one of the first to show that population density is an important driver of COVID-19 transmission, even in areas where residents rely more on private modes of transportation.”

Comment 3

3. It is not clear how the period of exponential growth was calibrated, and the reason for assuming the exponential growth period was one week prior to the second daily increases in cases.

Response 3

We agree with the reviewer that clarifying our method of determining the timing of the exponential growth period is very important. Our method actually assumed that the exponential growth period began one week prior to the second daily increase in cases. We have modified the text accordingly. In the same paragraph, we state the reasoning behind this assumption.

“The algorithm ensured that the virus had taken hold in the area and allowed a sufficient number of days to estimate the exponential growth rate, as R0 cannot be estimated accurately with sparse data, since it would be uncertain if the county was experiencing a sustained outbreak with community transmission.”

We understand that this method may have biases, and we added additional information in the Discussion section as a limitation (p.13)

“Furthermore, we utilized a number of assumptions based on previous findings to calibrate the exponential growth period, which ensured that the virus had taken hold and allowed a sufficient number of days and case counts to estimate exponential growth. There are potential for biases in our method… However, we implemented numerous ways to limit the biases. The exponential growth period was restricted to approximately 14 days at the start of the epidemic, where we would expect limited increases in testing and thus would not affect R0 substantially. Moreover, we plotted the calibrated exponential growth curves of all the counties included in our analysis, which gave us reasonable curves that approximated exponential growth for case and death data.”

Despite the limitations of our method, we believe there is value in using a relatively straightforward method to assess exponential growth period, that can potentially be used in other research studies of COVID-19 or future pandemics.

Comment 4

4. It is not clear how mixed linear models fit into the cross-sectional data structure, as those models are usually for multilevel/hierarchical, longitudinal, or correlated data. Does each county have repeated measurements? Are measurements made on clusters?

Response 4

We thank the reviewer for the valuable suggestion, and agree that clarification on the model specification in needed. As the reviewer correctly stated, linear mixed models are usually used for correlated data (non-independence), which can arise from a hierarchical structure. Our data has a hierarchical structure, as COVID-19 case numbers among counties within states are non-independent, potentially due to variable resource allocation and differing health systems across states. Thus, including a random intercept term for the states, and a fixed slope for the counties in the linear mixed models would be able to account for state-level correlation among counties. We have added text in the Methods section to further clarify this (p. 5):

“We then conducted a cross-sectional analysis using linear mixed models with random intercepts for each state and fixed slopes for the counties to assess the association of population density and R0. The linear mixed models allow the intercept to vary among states, which accounts for non-independence among counties within each state, potentially due to variable resource allocation and differing health systems across states.” 

Moreover, in order to clarify our model specification in the manuscript, we have included the linear mixed model equation (p. 6)

“The associated linear mixed model equation for model 4 is

 Y_ij=β_0j+β_1j l〖n(density1 unit increase)〗_ij+β_2j 〖% private transportation〗_ij+β_3j 〖median household income〗_ij +β_4j 〖% private transportation〗_ij*〖median household income〗_ij+e_ij

where β_0j=γ_00+u_0j, for the i-th county for the j-th state”

Comment 5

5. Figures seem to be poorly presented.

Response 5

We have changed the color scale of Figure 1 to improve the distinction between the R0 values. We also excluded Alaska and Hawaii for presentability, as is commonly done in spatial maps of the United States.

Moreover, we updated Figure 2 to add in grey points to represent the individual county population densities, which add valuable information to the Figure and increase presentability of the figure.

We have run our figures through the PACE tool, which ensures that all our figures meet PLOS One’s technical requirements.

Comment 6

6. It was possible that some NPIs were introduced in the initial outbreak stage of COVID-19 in some counties, such as public information campaigns. This would make the study assumption invalid.

Response 6

We thank the reviewer for noting the need to clarify this. There is the possibility that some NPIs were introduced in the initial outbreak stage of COVID-19 in some counties; however, if this was the case, then case counts and subsequently R0 would even higher than we calculated, and thus our associations of density and R0 was underestimated. Moreover, we plotted the calibrated exponential growth curves of all the counties included in our analysis, which gave us reasonable curves that approximated exponential growth for case and death data. This gives further validity to our calibration method of the exponential growth period (S1_Appendix).

We have added this as a we have added this as a potential limitation in our study, as well as the measures we put in place to limit these potential biases (p. 13).

“There are potential for biases in our method; for example, there is the possibility that some NPIs were introduced in the initial outbreak stage of COVID-19 in some counties; however, if this was the case, then case counts and subsequently R0 would even higher than we calculated, and thus our associations of density and R0 was underestimated. However, we implemented numerous ways to limit the biases. The exponential growth period was restricted to approximately 14 days at the start of the epidemic, where we would expect limited increases in testing and thus would not affect R0 substantially. Moreover, we plotted the calibrated exponential growth curves of all the counties included in our analysis, which gave us reasonable curves that approximated exponential growth for case and death data.”

Comment 7

7. Authors are advised to provide explanations for insignificance of the coefficient estimate of population density in Model 4.

Response 7

We thank the reviewer for the valuable suggestion for the insignificance of the coefficient estimate of population density in Model 4. Since the interaction between population density and transportation was non-significant, this demonstrates that density-dependence is not differential across transportation accessibility, and thus Model 4 results should not be interpreted. We attempted to make this clearer in the Discussion section (p. 11, last paragraph)

“However, we did not see that density-dependence is differential across transportation accessibility, as demonstrated by the non-significant interaction of population density and transportation. Therefore, model 4 should be not be interpreted, since the interaction does not contribute to the model and only serves to decrease the precision (larger confidence intervals) of the measures of association of interest.”

---

## [Decision Letter · Decision Letter 1]

16 Mar 2021

Population density and basic reproductive number of COVID-19 across United States counties

PONE-D-20-40744R1

Dear Dr. Nichols,

We’re pleased to inform you that your manuscript has been judged scientifically suitable for publication and will be formally accepted for publication once it meets all outstanding technical requirements.

Kind regards,

Martial L Ndeffo Mbah, Ph.D

Academic Editor

PLOS ONE

Additional Editor Comments (optional):

Reviewers' comments:

Reviewer's Responses to Questions

**Comments to the Author**

1. If the authors have adequately addressed your comments raised in a previous round of review and you feel that this manuscript is now acceptable for publication, you may indicate that here to bypass the “Comments to the Author” section, enter your conflict of interest statement in the “Confidential to Editor” section, and submit your "Accept" recommendation.

Reviewer #1: All comments have been addressed

Reviewer #2: All comments have been addressed

2. Is the manuscript technically sound, and do the data support the conclusions?

Reviewer #1: Yes

Reviewer #2: Yes

3. Has the statistical analysis been performed appropriately and rigorously? 

Reviewer #1: Yes

Reviewer #2: Yes

4. Have the authors made all data underlying the findings in their manuscript fully available?

Reviewer #1: Yes

Reviewer #2: Yes

5. Is the manuscript presented in an intelligible fashion and written in standard English?

Reviewer #1: Yes

Reviewer #2: Yes

6. Review Comments to the Author

Reviewer #1: Thank you for addressing my comments. While R0 is a complex measure to calculate, and is easily misrepresented, misinterpreted, and misapplied, it is a fundamental metric for the study of infectious disease. This research, therefore, adds to the scientific literature on COVID-19 in a meaningful way.

Reviewer #2: The authors have addressed issues raised in the reviewer report. I hope the readers will benefit from this work.

7. PLOS authors have the option to publish the peer review history of their article (what does this mean?). If published, this will include your full peer review and any attached files.

Reviewer #1: No

Reviewer #2: No

---

## [Editor Report · Acceptance letter]

31 Mar 2021

PONE-D-20-40744R1 

Population density and basic reproductive number of COVID-19 across United States counties 

Dear Dr. Nichols:

I'm pleased to inform you that your manuscript has been deemed suitable for publication in PLOS ONE. Congratulations! Your manuscript is now with our production department. 

Kind regards, 

on behalf of

Dr. Martial L Ndeffo Mbah 

Academic Editor

PLOS ONE